# Learning to Infer Graphics Programs from Hand-Drawn Images

**Kevin Ellis**
MIT
ellisk@mit.edu

**Daniel Ritchie**
Brown University
daniel_ritchie@brown.edu

**Armando Solar-Lezama**
MIT
asolar@csail.mit.edu

**Joshua B. Tenenbaum**
MIT
jbt@mit.edu

## Abstract

We introduce a model that learns to convert simple hand drawings into graphics programs written in a subset of LaTeX. The model combines techniques from deep learning and program synthesis. We learn a convolutional neural network that proposes plausible drawing primitives that explain an image. These drawing primitives are a specification (spec) of what the graphics program needs to draw. We learn a model that uses program synthesis techniques to recover a graphics program from that spec. These programs have constructs like variable bindings, iterative loops, or simple kinds of conditionals. With a graphics program in hand, we can correct errors made by the deep network and extrapolate drawings.

## 1 Introduction

Human vision is rich – we infer shape, objects, parts of objects, and relations between objects – and vision is also abstract: we can perceive the radial symmetry of a spiral staircase, the iterated repetition in the Ising model, see the forest for the trees, and also the recursion within the trees. How could we build an agent with similar visual inference abilities? As a small step in this direction, we cast this problem as program learning, and take as our goal to learn high–level graphics programs from simple 2D drawings. The graphics programs we consider make figures like those found in machine learning papers (Fig. 1), and capture high-level features like symmetry, repetition, and reuse of structure.

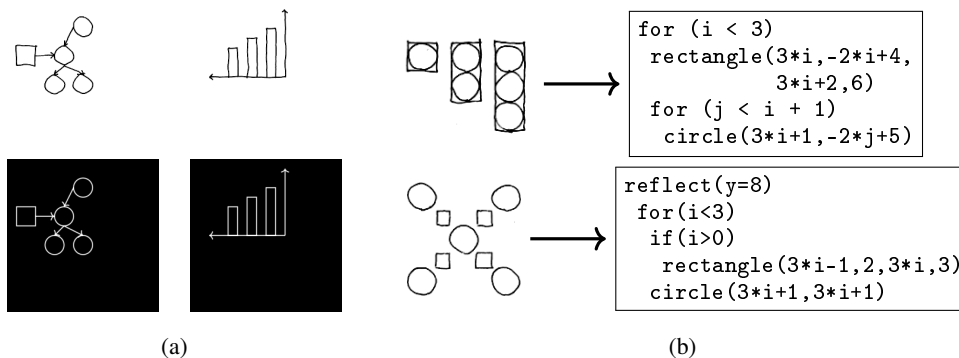

(a)                                                    (b)

Figure 1: (a): Model learns to convert hand drawings (top) into LaTeX (rendered below). (b) Learns to synthesize high-level graphics program from hand drawing.

The key observation behind our work is that going from pixels to programs involves two distinct steps, each requiring different technical approaches. The first step involves inferring what objects make up an image – for diagrams, these are things like as rectangles, lines and arrows. The second step involves identifying the higher-level visual concepts that describe how the objects were drawn. In Fig. 1(b), it means identifying a pattern in how the circles and rectangles are being drawn that is best described with two nested loops, and which can easily be extrapolated to a bigger diagram.

This two-step factoring can be framed as probabilistic inference in a generative model where a latent program is executed to produce a set of drawing commands, which are then rendered to form an image (Fig. 2). We refer to this set of drawing commands as a **specification (spec)** because it specifies what the graphics program drew while lacking the high-level structure determining how the program decided to draw it. We infer a spec from an image using stochastic search (Sequential Monte Carlo) and infer a program from a spec using constraint-based program synthesis [1] – synthesizing structures like symmetries, loops, or conditionals. In practice, both stochastic search and program synthesis are prohibitively slow, and so we learn models that accelerate inference for both programs and specs, in the spirit of "amortized inference" [2], training a neural network to amortize the cost of inferring specs from images and using a variant of Bias–Optimal Search [3] to amortize the cost of synthesizing programs from specs.

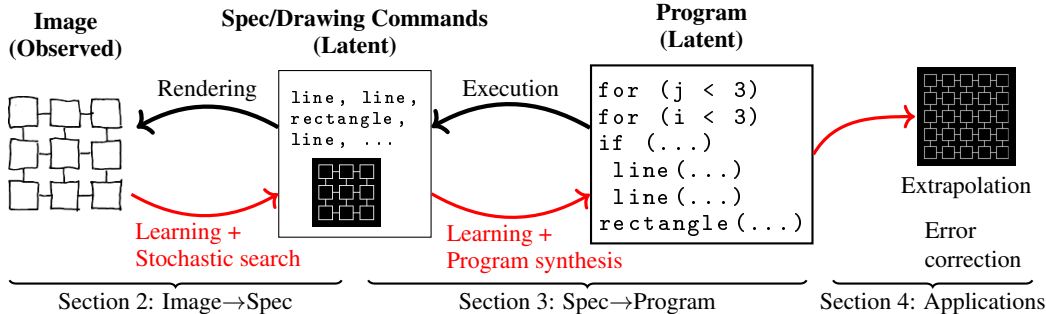

Figure 2: Black arrows: Top–down generative model; Program→Spec→Image. Red arrows: Bottom–up inference procedure. **Bold:** Random variables (image/spec/program)

The new contributions of this work are (1) a working model that can infer high-level symbolic programs from perceptual input, and (2) a technique for using learning to amortize the cost of program synthesis, described in Section 3.1.

## 2    Neural architecture for inferring specs

We developed a deep network architecture for efficiently inferring a spec, $S$, from a hand-drawn image, $I$. Our model combines ideas from Neurally-Guided Procedural Models [4] and Attend-Infer-Repeat [5], but we wish to emphasize that one could use many different approaches from the computer vision toolkit to parse an image in to primitive drawing commands (in our terminology, a "spec") [6]. Our network constructs the spec one drawing command at a time, conditioned on what it has drawn so far (Fig. 3). We first pass a $256 \times 256$ target image and a rendering of the drawing commands so far (encoded as a two-channel image) to a convolutional network. Given the features extracted by the convnet, a multilayer perceptron then predicts a distribution over the next drawing command to execute (see Tbl. 1). We also use a differentiable attention mechanism (Spatial Transformer Networks: [7]) to let the model attend to different regions of the image while predicting drawing commands. We currently constrain coordinates to lie on a discrete $16 \times 16$ grid, but the grid could be made arbitrarily fine.

We trained our network by sampling specs $S$ and target images $I$ for randomly generated scenes[1] and maximizing $\mathbb{P}_\theta[S|I]$, the likelihood of $S$ given $I$, with respect to model parameters $\theta$, by gradient ascent. We trained on $10^5$ scenes, which takes a day on an Nvidia TitanX GPU. Supplement Section 1 gives the full details of the architecture and training of this network.

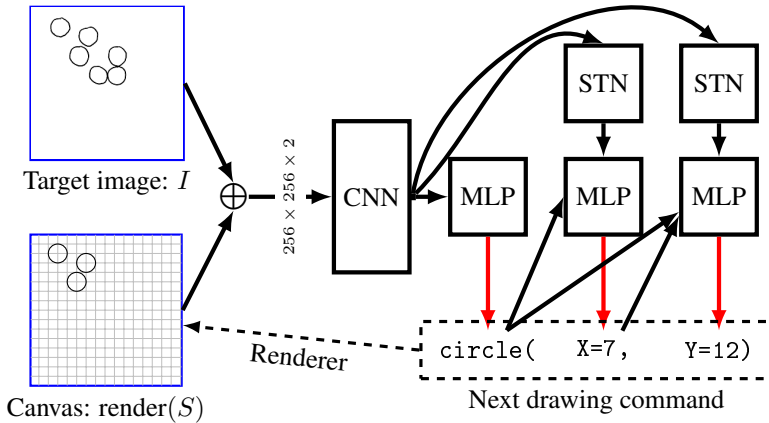

Target image: $I$

$256 \times 256 \times 2$

Renderer

Canvas: render($S$)

circle(    X=7,        Y=12)

Next drawing command

Figure 3: Neural architecture for inferring specs from images. Blue: network inputs. Black: network operations. Red: draws from a multinomial. `Typewriter font`: network outputs. Renders on a $16 \times 16$ grid, shown in gray. STN: differentiable attention mechanism [7].

Table 1: Primitive drawing commands currently supported by our model.

| | |
|---|---|
| $\texttt{circle}(x, y)$ | Circle at $(x, y)$ |
| $\texttt{rectangle}(x_1, y_1, x_2, y_2)$ | Rectangle with corners at $(x_1, y_1)$ & $(x_2, y_2)$ |
| $\texttt{line}(x_1, y_1, x_2, y_2,$ | Line from $(x_1, y_1)$ to $(x_2, y_2)$, |
| $\quad \text{arrow} \in \{0, 1\}, \text{dashed} \in \{0, 1\})$ | optionally with an arrow and/or dashed |
| $\texttt{STOP}$ | Finishes spec inference |

Our network can "derender" random synthetic images by doing a beam search to recover specs maximizing $\mathbb{P}_\theta[S|I]$. But, if the network predicts an incorrect drawing command, it has no way of recovering from that error. For added robustness we treat the network outputs as proposals for a Sequential Monte Carlo (SMC) sampling scheme [8]. Our SMC sampler draws samples from the distribution $\propto L(I|\text{render}(S))\mathbb{P}_\theta[S|I]$, where $L(\cdot|\cdot)$ uses the pixel-wise distance between two images as a proxy for a likelihood. Here, the network is learning a proposal distribution to amortize the cost of inverting a generative model (the renderer) [2].

**Experiment 1: Figure 4.** To evaluate which components of the model are necessary to parse complicated scenes, we compared the neural network with SMC against the neural network by itself (i.e., w/ beam search) or SMC by itself. Only the combination of the two passes a critical test of generalization: when trained on images with $\leq 12$ objects, it successfully parses scenes with many more objects than the training data. We compare with a baseline that produces the spec in one shot by using the CNN to extract features of the input which are passed to an LSTM which finally predicts the spec token-by-token (LSTM in Fig. 4). This architecture is used in several successful neural models of image captioning (e.g., [9]), but, for this domain, cannot parse cluttered scenes with many objects.

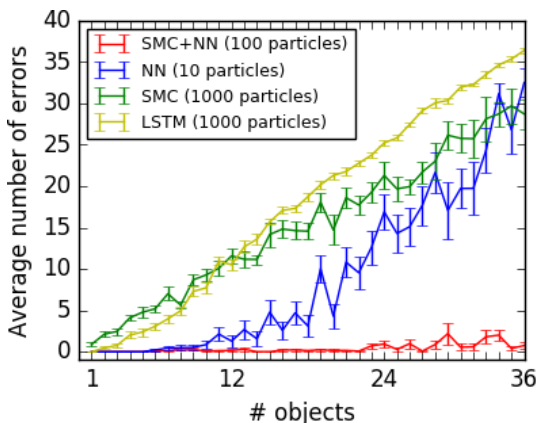

Figure 4: Parsing LaTeX output after training on diagrams with $\leq 12$ objects. Out-of-sample generalization: Model generalizes to scenes with many more objects ($\approx$ at ceiling when tested on twice as many objects as were in the training data). Neither SMC nor the neural network are sufficient on their own. # particles varies by model: we compare the models *with equal runtime* ($\approx 1$ sec/object). Average number of errors is (# incorrect drawing commands predicted by model)+(# correct commands that were not predicted by model).

## 2.1 Generalizing to real hand drawings

We trained the model to generalize to hand drawings by introducing noise into the renderings of the training target images, where the noise process mimics the kinds of variations found in hand drawings. While our neurally-guided SMC procedure used pixel-wise distance as a surrogate for a likelihood function ($L(\cdot|\cdot)$ in Sec. 2), pixel-wise distance fares poorly on hand drawings, which never exactly match the model's renders. So, for hand drawings, we learn a surrogate likelihood function, $L_{\text{learned}}(\cdot|\cdot)$. The density $L_{\text{learned}}(\cdot|\cdot)$ is predicted by a convolutional network that we train to predict the distance between two specs conditioned upon their renderings. We train $L_{\text{learned}}(\cdot|\cdot)$ to approximate the symmetric difference, which is the number of drawing commands by which two specs differ:

$$-\log L_{\text{learned}}(\text{render}(S_1)|\text{render}(S_2)) \approx |S_1 - S_2| + |S_2 - S_1| \qquad (1)$$

Supplement Section 2 explains the architecture and training of $L_{\text{learned}}$.

**Experiment 2: Figures 5–7.** We evaluated, but did not train, our system on 100 real hand-drawn figures; see Fig. 5–6. These were drawn carefully but not perfectly with the aid of graph paper. For each drawing we annotated a ground truth spec and had the neurally guided SMC sampler produce $10^3$ samples. For 63% of the drawings, the Top-1 most likely sample exactly matches the ground truth; with more samples, the model finds specs that are closer to the ground truth annotation (Fig. 7). We will show that the program synthesizer corrects some of these small errors (Sec. 4.1).

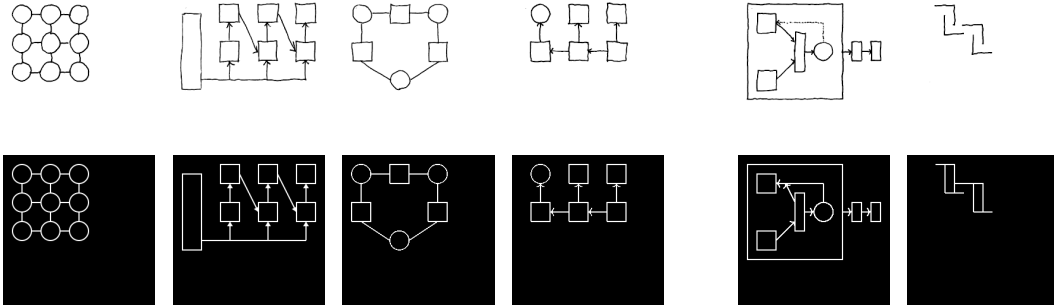

Figure 5: Left to right: Ising model, recurrent network architecture, figure from a deep learning textbook [10], graphical model

Figure 6: Near misses. Rightmost: illusory contours (note: no SMC in rightmost)

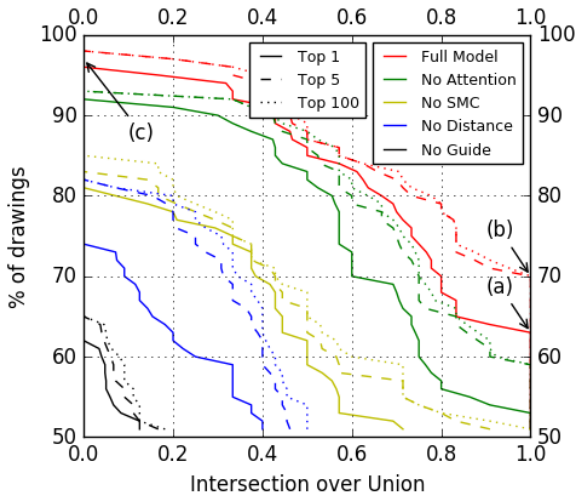

Figure 7: How close are the model's outputs to the ground truth on hand drawings, as we consider larger sets of samples (1, 5, 100)? Distance to ground truth measured by the intersection over union (IoU) of predicted spec vs. ground truth spec: IoU of sets (specs) $A$ and $B$ is $|A \cap B|/|A \cup B|$. (a) for 63% of drawings the model's top prediction is exactly correct; (b) for 70% of drawings the ground truth is in the top 5 model predictions; (c) for 4% of drawings all of the model outputs have no overlap with the ground truth. Red: the full model. Other colors: lesioned versions of our model.

## 3 Synthesizing graphics programs from specs

Although the spec describes the contents of a scene, it does not encode higher-level features of an image such as repeated motifs or symmetries, which are more naturally captured by a graphics program. We seek to synthesize graphics programs from their specs.

We constrain the space of programs by writing down a context free grammar over programs – what in the program languages community is called a Domain Specific Language (DSL) [11]. Our DSL (Tbl. 2) encodes prior knowledge of what graphics programs tend to look like.

Table 2: Grammar over graphics programs. We allow loops (`for`) with conditionals (`if`), vertical/horizontal reflections (`reflect`), variables (Var) and affine transformations ($\mathbb{Z}\times$Var$+\mathbb{Z}$).

| | |
|---:|:---|
| Program→ | Statement; $\cdots$; Statement |
| Statement→ | `circle`(Expression,Expression) |
| Statement→ | `rectangle`(Expression,Expression,Expression,Expression) |
| Statement→ | `line`(Expression,Expression,Expression,Expression,Boolean,Boolean) |
| Statement→ | `for`($0 \leq$ Var $<$ Expression) { `if` (Var $> 0$) { Program }; Program } |
| Statement→ | `reflect`(Axis) { Program } |
| Expression→ | $\mathbb{Z}\times$Var$+\mathbb{Z}$ |
| Axis→ | `X = `$\mathbb{Z}$` \| Y = `$\mathbb{Z}$ |
| $\mathbb{Z} \rightarrow$ | an integer |

Given the DSL and a spec $S$, we want a program that both satisfies $S$ and, at the same time, is the "best" explanation of $S$. For example, we might prefer more general programs or, in the spirit of Occam's razor, prefer shorter programs. We wrap these intuitions up into a cost function over programs, and seek the minimum cost program consistent with $S$:

$$\text{program}(S) = \underset{p\in\text{DSL}}{\arg\max}\ \mathbb{1}\left[p \text{ consistent w/ } S\right]\exp\left(-\text{cost}(p)\right) \qquad (2)$$

We define the cost of a program to be the number of Statement's it contains (Tbl. 2). We also penalize using many different numerical constants; see Supplement Section 3. Returning to the generative model in Fig. 2, this setup is the same as saying that the prior probability of a program $p$ is $\propto \exp\left(-\text{cost}(p)\right)$ and the likelihood of a spec $S$ given a program $p$ is $\mathbb{1}[p \text{ consistent w/ } S]$.

The constrained optimization problem in Eq. 2 is intractable in general, but there exist efficient-in-practice tools for finding exact solutions to such program synthesis problems. We use the state-of-the-art Sketch tool [1]. Sketch takes as input a space of programs, along with a specification of the program's behavior and optionally a cost function. It translates the synthesis problem into a constraint satisfaction problem and then uses a SAT solver to find a minimum-cost program satisfying the specification. Sketch requires a *finite program space*, which here means that the depth of the program syntax tree is bounded (we set the bound to 3), but has the guarantee that it always eventually finds a globally optimal solution. In exchange for this optimality guarantee it comes with no guarantees on runtime. For our domain synthesis times vary from minutes to hours, with 27% of the drawings timing out the synthesizer after 1 hour. Tbl. 3 shows programs recovered by our system. A main impediment to our use of these general techniques is the prohibitively high cost of searching for programs. We next describe how to learn to synthesize programs much faster (Sec. 3.1), timing out on 2% of the drawings and solving 58% of problems within a minute.

## 3.1 Learning a search policy for synthesizing programs

We want to leverage powerful, domain-general techniques from the program synthesis community, but make them much faster by learning a domain-specific **search policy**. A search policy poses search problems like those in Eq. 2, but also offers additional constraints on the structure of the program (Tbl. 4). For example, a policy might decide to first try searching over small programs before searching over large programs, or decide to prioritize searching over programs that have loops.

A search policy $\pi_\theta(\sigma|S)$ takes as input a spec $S$ and predicts a distribution over synthesis problems, each of which is written $\sigma$ and corresponds to a set of possible programs to search over (so $\sigma \subseteq$ DSL). Good policies will prefer tractable program spaces, so that the search procedure will terminate early, but should also prefer program spaces likely to contain programs that concisely explain the data. These two desiderata are in tension: tractable synthesis problems involve searching over smaller spaces, but smaller spaces are less likely to contain good programs. Our goal now is to find the parameters of the policy, written $\theta$, that best navigate this trade-off.

Given a search policy, what is the best way of using it to quickly find minimum cost programs? We use a bias-optimal search algorithm (c.f. Schmidhuber 2004 [3]):

Table 3: Drawings (left), their specs (middle left), and programs synthesized from those specs (middle right). Compared to the specs the programs are more compressive (right: programs have fewer lines than specs) and automatically group together related drawing commands. Note the nested loops and conditionals in the Ising model, combination of symmetry and iteration in the bottom figure, affine transformations in the top figure, and the complicated program in the second figure to bottom.

| Drawing | Spec | Program | Compression factor |
|---|---|---|---|
| 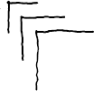 | `Line(2,15, 4,15)`<br>`Line(4,9, 4,13)`<br>`Line(3,11, 3,14)`<br>`Line(2,13, 2,15)`<br>`Line(3,14, 6,14)`<br>`Line(4,13, 8,13)` | `for(i<3)`<br>` line(i,-1*i+6,`<br>`     2*i+2,-1*i+6)`<br>` line(i,-2*i+4,i,-1*i+6)` | $\frac{6}{3} = 2\text{x}$ |
|  | `Circle(5,8)`<br>`Circle(2,8)`<br>`Circle(8,11)`<br>`Line(2,9, 2,10)`<br>`Circle(8,8)`<br>`Line(3,8, 4,8)`<br>`Line(3,11, 4,11)`<br>*... etc. ...; 21 lines* | `for(i<3)`<br>` for(j<3)`<br>`  if(j>0)`<br>`   line(-3*j+8,-3*i+7,`<br>`        -3*j+9,-3*i+7)`<br>`   line(-3*i+7,-3*j+8,`<br>`        -3*i+7,-3*j+9)`<br>`   circle(-3*j+7,-3*i+7)` | $\frac{21}{6} = 3.5\text{x}$ |
|  | `Rectangle(1,10,3,11)`<br>`Rectangle(1,12,3,13)`<br>`Rectangle(4,8,6,9)`<br>`Rectangle(4,10,6,11)`<br>*... etc. ...; 16 lines* | `for(i<4)`<br>` for(j<4)`<br>`  rectangle(-3*i+9,-2*j+6,`<br>`            -3*i+11,-2*j+7)` | $\frac{16}{3} = 5.3\text{x}$ |
|  | `Line(11,14,13,14,arrow)`<br>`Circle(10,10)`<br>`Line(10,13,10,11,arrow)`<br>`Circle(6,10)`<br>*... etc. ...; 15 lines* | `for(i<4)`<br>` line(-4*i+13,4,-4*i+13,2,arrow)`<br>` for(j<3)`<br>`  if(j>0)`<br>`   circle(-4*i+13,4*j+-3)`<br>`  line(-4*j+10,5,-4*j+12,5,`<br>`       arrow)` | $\frac{15}{6} = 2.5\text{x}$ |
|  | `Line(3,10,3,14,arrow)`<br>`Rectangle(11,8,15,10)`<br>`Rectangle(11,14,15,15)`<br>`Line(13,10,13,14,arrow)`<br>*... etc. ...; 16 lines* | `for(i<3)`<br>` line(7,1,5*i+2,3,arrow)`<br>` for(j<i+1)`<br>`  if(j>0)`<br>`   line(5*j-1,9,5*i,5,arrow)`<br>`  line(5*j+2,5,5*j+2,9,arrow)`<br>` rectangle(5*i,3,5*i+4,5)`<br>` rectangle(5*i,9,5*i+4,10)`<br>`rectangle(2,0,12,1)` | $\frac{16}{9} = 1.8\text{x}$ |
|  | `Circle(2,8)`<br>`Rectangle(6,9, 7,10)`<br>`Circle(8,8)`<br>`Rectangle(6,12, 7,13)`<br>`Rectangle(3,9, 4,10)`<br>*... etc. ...; 9 lines* | `reflect(y=8)`<br>` for(i<3)`<br>`  if(i>0)`<br>`   rectangle(3*i-1,2,3*i,3)`<br>`  circle(3*i+1,3*i+1)` | $\frac{9}{5} = 1.8\text{x}$ |

**Definition: Bias-optimality.** A search algorithm is *n-bias optimal* with respect to a distribution $\mathbb{P}_{\text{bias}}[\cdot]$ if it is guaranteed to find a solution in $\sigma$ after searching for at least time $n \times \frac{t(\sigma)}{\mathbb{P}_{\text{bias}}[\sigma]}$, where $t(\sigma)$ is the time it takes to verify that $\sigma$ contains a solution to the search problem.

Bias-optimal search over program spaces is known as **Levin Search** [12]; an example of a 1-bias optimal search algorithm is an ideal time-sharing system that allocates $\mathbb{P}_{\text{bias}}[\sigma]$ of its time to trying $\sigma$. We construct a 1-bias optimal search algorithm by identifying $\mathbb{P}_{\text{bias}}[\sigma] = \pi_\theta(\sigma|S)$ and $t(\sigma) = t(\sigma|S)$, where $t(\sigma|S)$ is how long the synthesizer takes to search $\sigma$ for a program for $S$. Intuitively, this means that the search algorithm explores the entire program space, but spends most of its time in the regions of the space that the policy judges to be most promising. Concretely, this means that we run many different program searches *in parallel* (i.e., run in parallel different instances of the synthesizer, one for each $\sigma$), but to allocate compute time to a $\sigma$ in proportion to $\pi_\theta(\sigma|S)$.

Now in theory any $\pi_\theta(\cdot|\cdot)$ is a bias-optimal searcher. But the actual runtime of the algorithm depends strongly upon the bias $\mathbb{P}_{\text{bias}}[\cdot]$. Our new approach is to learn $\mathbb{P}_{\text{bias}}[\cdot]$ by picking the policy minimizing the expected bias-optimal time to solve a training corpus, $\mathcal{D}$, of graphics program synthesis problems:

$$\text{Loss}(\theta; \mathcal{D}) = \mathbb{E}_{S \sim \mathcal{D}} \left[ \min_{\sigma \in \text{BEST}(S)} \frac{t(\sigma|S)}{\pi_\theta(\sigma|S)} \right] + \lambda \|\theta\|_2^2 \tag{3}$$

where $\sigma \in \text{BEST}(S)$ if a minimum cost program for $S$ is in $\sigma$.

To generate a training corpus for learning a policy, we synthesized minimum cost programs for each drawing and for each $\sigma$, then minimized 3 using gradient descent while annealing a softened minimum to the hard minimization equation 3. Because we want to learn a policy from only 100 drawings, we parameterize $\pi$ with a low-capacity bilinear model with only 96 real-valued parameters. Supplement Section 4 further details the parameterization and training of the policy.

**Experiment 3: Table 5; Figure 8; Supplement Section 4.** We compare synthesis times for our learned search policy with 4 alternatives: *Sketch*, which poses the entire problem wholesale to the Sketch program synthesizer; *DC*, a DeepCoder–style model that learns to predict which program components (loops, reflections) are likely to be useful [13]; *End–to-End*, which trains a recurrent neural network to regress directly from images to programs; and an *Oracle*, a policy which always picks the quickest to search $\sigma$ also containing a minimum cost program. Our approach improves upon Sketch by itself, and comes close to the Oracle's performance. One could never construct this Oracle, because the agent does not know ahead of time which $\sigma$'s contain minimum cost programs nor does it know how long each $\sigma$ will take to search. With this learned policy in hand we can synthesize 58% of programs within a minute.

Table 4: Parameterization of different ways of posing the program synthesis problem. The policy learns to choose parameters likely to quickly yield a minimal cost program.

| Parameter | Description | Range |
|---|---|---|
| Loops? | Is the program allowed to loop? | $\{\text{True}, \text{False}\}$ |
| Reflects? | Is the program allowed to have reflections? | $\{\text{True}, \text{False}\}$ |
| Incremental? | Solve the problem piece-by-piece or all at once? | $\{\text{True}, \text{False}\}$ |
| Maximum depth | Bound on the depth of the program syntax tree | $\{1, 2, 3\}$ |

| Model | Median search time | Timeouts (1 hr) |
|---|---|---|
| Sketch | 274 sec | 27% |
| DC | 187 sec | 2% |
| End–to–End | 63 sec | 94% |
| Oracle | 6 sec | 2% |
| Ours | 28 sec | 2% |

Table 5: Time to synthesize a minimum cost program. Sketch: out-of-the-box performance of Sketch [1]. DC: Deep–Coder style baseline that predicts program components, trained like [13]. End–to–End: neural net trained to regress directly from images to programs, which fails to find valid programs 94% of the time. Oracle: upper bounds the performance of any bias–optimal search policy. Ours: evaluated w/ 20-fold cross validation.

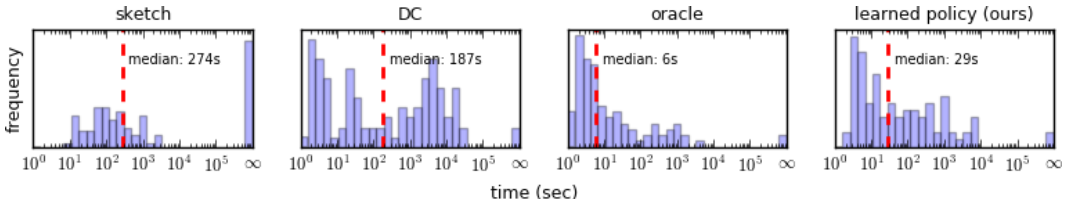

Figure 8: Time to synthesize a minimum cost program (compare w/ Table 5). End–to–End: not shown because it times out on 96% of drawings, and has its median time (63s) calculated only on non-timeouts, wheras the other comparisons include timeouts in their median calculation. $\infty$ = timeout. Red dashed line is median time.

# 4 Applications of graphics program synthesis

## 4.1 Correcting errors made by the neural network

The program synthesizer corrects errors made by the neural network by favoring specs which lead to more concise or general programs. For example, figures with perfectly aligned objects are preferable, and precise alignment lends itself to short programs. Concretely, we run the program synthesizer on the Top-$k$ most likely specs output by the neurally guided sampler. Then, the system reranks the Top-$k$ by the prior probability of their programs. The prior probability of a program is learned by optimizing the parameters of the prior so as to maximize the likelihood of the ground truth specs; see supplement for details. But, this procedure can only correct errors when a correct spec is in the Top-$k$. Our sampler could only do better on 7/100 drawings by looking at the Top-100 samples (see Fig. 7), precluding a statistically significant analysis of how much learning a prior over programs could help correct errors. But, learning this prior does sometimes help correct mistakes made by the neural network; see Fig. 9 for a representative example of the kinds of corrections that it makes. See Supplement Section 5 for details.

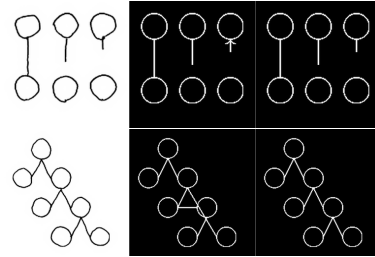

Figure 9: Left: hand drawings. Center: interpretations favored by the deep network. Right: interpretations favored after learning a prior over programs. The prior favors simpler programs, thus (top) continuing the pattern of not having an arrow is preferred, or (bottom) continuing the "binary search tree" is preferred.

## 4.2 Extrapolating figures

Having access to the source code of a graphics program facilitates coherent, high-level image editing. For example, we could change all of the circles to squares or make all of the lines be dashed, or we can (automatically) extrapolate figures by increasing the number of times that loops are executed. Extrapolating repetitive visuals patterns comes naturally to humans, and is a practical application: imagine hand drawing a repetitive graphical model structure and having our system automatically induce and extend the pattern. Fig. 10 shows extrapolations produced by our system.

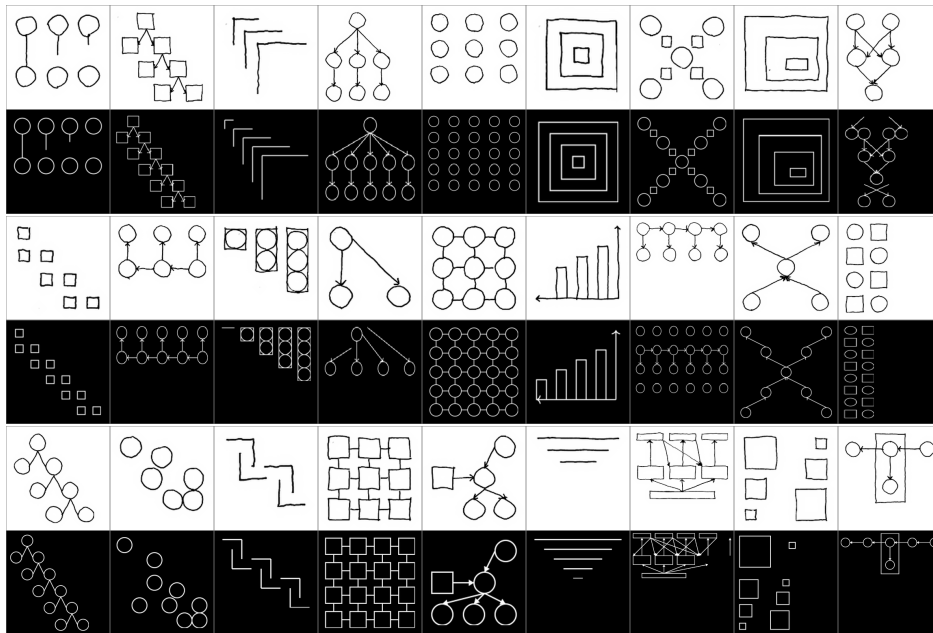

Figure 10: Top, white: drawings. Bottom, black: extrapolations automatically produced by our system.

# 5   Related work

**Program Induction:** Our approach to learning to search for programs draws theoretical under-pinnings from Levin search [12, 14] and Schmidhuber's OOPS model [3]. DeepCoder [13] is a recent model which, like ours, learns to predict likely program components. Our work differs by identifying and modeling the trade-off between tractability and probability of success. TerpreT [15] systematically compares constraint-based program synthesis techniques against gradient-based search methods, like those used to train Differentiable Neural Computers [16]. The TerpreT experiments motivate our use of constraint-based techniques. Neurally Guided Deductive Search (NGDS: [17]) is a recent neurosymbolic approach; combining our work with ideas from NGDS could be promising.

**Deep Learning:** Our neural network combines the architectural ideas of Attend-Infer-Repeat [5] – which learns to decompose an image into its constituent objects – with the training regime and SMC inference of Neurally Guided Procedural Modeling [4] – which learns to control procedural graphics programs. The very recent SPIRAL [18] system learns to infer procedures for controlling a 'pen' to derender highly diverse natural images, complementing our focus here on more abstract procedures but less natural images. IM2LATEX [19] and pix2code [20] are recent works that derender LATEX equations and GUIs, respectively, both recovering a markup-like representation. Our goal is to go from noisy input to a high-level program, which goes beyond markup languages by supporting programming constructs like loops and conditionals.

**Hand-drawn sketches:** Sketch-n-Sketch is a bi-directional editing system where direct manipulations to a program's output automatically propagate to the program source code [21]. This work compliments our own: programs produced by our method could be provided to a Sketch-n-Sketch-like system as a starting point for further editing. Other systems in the computer graphics literature convert sketches to procedural representations, e.g. using a convolutional network to match a sketch to the output of a parametric 3D modeling system in [22] or supporting interactive sketch-based instantiation of procedural primitives in [23] In contrast, we seek to automatically infer a programmatic representation capturing higher-level visual patterns. The CogSketch system [24] also aims to have a high-level understanding of hand-drawn figures. Their goal is cognitive modeling, whereas we are interested in building an automated AI application.

# 6   Contributions

We have presented a system for inferring graphics programs which generate LATEX-style figures from hand-drawn images using a combination of learning, stochastic search, and program synthesis. In the near future, we believe it will be possible to produce professional-looking figures just by drawing them and then letting an AI write the code. More generally, we believe the problem of inferring visual programs is a promising direction for research in machine perception.

### Acknowledgments

We are grateful for advice from Will Grathwohl and Jiajun Wu on the neural architecture, and for funding from NSF GRFP, NSF Award #1753684, the MUSE program (DARPA grant FA8750-14-2-0242), and AFOSR award FA9550-16-1-0012. This material is based upon work supported by the Center for Brains, Minds and Machines (CBMM), funded by NSF STC award CCF-1231216.

### Code, data, and drafts

A longer version of this paper is available at `https://arxiv.org/abs/1707.09627`. The code and data are available at `https://github.com/ellisk42/TikZ`.

## Footnotes

[1]Because rendering ignores ordering we put the drawing commands into a canonical order

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
