[Supplementary Material]

# Supplement to: Learning to Infer Graphics Programs from Hand-Drawn Images

**Kevin Ellis**
MIT
ellisk@mit.edu

**Daniel Ritchie**
Brown University
daniel_ritchie@brown.edu

**Armando Solar-Lezama**
MIT
asolar@csail.mit.edu

**Joshua B. Tenenbaum**
MIT
jbt@mit.edu

## 1 Neural network architecture and training

### 1.1 High-level overview

For the model in Fig. 1, the distribution over the next drawing command factorizes as:

$$\mathbb{P}_\theta[t_1 t_2 \cdots t_K | I, S] = \prod_{k=1}^{K} \mathbb{P}_\theta \left[ t_k | a_\theta \left( f_\theta(I, \text{render}(S)) | \{t_j\}_{j=1}^{k-1} \right), \{t_j\}_{j=1}^{k-1} \right] \tag{1}$$

where $t_1 t_2 \cdots t_K$ are the tokens in the drawing command, $I$ is the target image, $S$ is a spec, $\theta$ are the parameters of the neural network, $f_\theta(\cdot, \cdot)$ is the image feature extractor (convolutional network), and $a_\theta(\cdot|\cdot)$ is an attention mechanism. The distribution over specs factorizes as:

$$\mathbb{P}_\theta[S|I] = \prod_{n=1}^{|S|} \mathbb{P}_\theta[S_n | I, S_{1:(n-1)}] \times \mathbb{P}_\theta[\texttt{STOP}|I, S] \tag{2}$$

where $|S|$ is the length of spec $S$, the subscripts on $S$ index drawing commands within the spec (so $S_n$ is a sequence of tokens: $t_1 t_2 \cdots t_K$), and the STOP token is emitted by the network to signal that the spec explains the image.

### 1.2 Convolutional network

The convolutional network takes as input 2 $256 \times 256$ images represented as a $2 \times 256 \times 256$ volume. These are passed through two layers of convolutions separated by ReLU nonlinearities and max pooling:

- Layer 1: 20 $8 \times 8$ convolutions, 2 $16 \times 4$ convolutions, 2 $4 \times 16$ convolutions. Followed by $8 \times 8$ pooling with a stride size of 4.
- Layer 2: 10 $8 \times 8$ convolutions. Followed by $4 \times 4$ pooling with a stride size of 4.

### 1.3 Autoregressive decoding of drawing commands

Given the image features $f$, we predict the first token (i.e., the name of the drawing command: `circle`, `rectangle`, `line`, or `STOP`) using logistic regression:

$$\mathbb{P}[t_1] \propto \exp\left(W_{t_1} f + b_{t_1}\right) \tag{3}$$

where $W_{t_1}$ is a learned weight matrix and $b_{t_1}$ is a learned bias vector.

Figure 1: Our neural architecture for inferring the spec of a graphics program from its output. Blue: network inputs. Black: network operations. Red: samples from a multinomial. `Typewriter font`: network outputs. Renders snapped to a $16 \times 16$ grid, illustrated in gray. STN (spatial transformer network) is a differentiable attention mechanism [7].

Given an attention mechanism $a(\cdot|\cdot)$, subsequent tokens are predicted as:

$$\mathbb{P}[t_n|t_{1:(n-1)}] \propto \mathrm{MLP}_{t_1,n}(a(f|t_{1:(n-1)}) \oplus \bigoplus_{j<n} \mathrm{oneHot}(t_j)) \tag{4}$$

Thus each token of each drawing primitive has its own learned MLP. For predicting the coordinates of lines we found that using 32 hidden nodes with sigmoid activations worked well; for other tokens the MLP's are just logistic regression (no hidden nodes).

We use Spatial Transformer Networks [7] as our attention mechanism. The parameters of the spatial transform are predicted on the basis of previously predicted tokens. For example, in order to decide where to focus our attention when predicting the $y$ coordinate of a circle, we condition upon both the identity of the drawing command (`circle`) and upon the value of the previously predicted $x$ coordinate:

$$a(f|t_{1:(n-1)}) = \mathrm{AffineTransform}(f, \mathrm{MLP}_{t_1,n}(\bigoplus_{j<n} \mathrm{oneHot}(t_j))) \tag{5}$$

So, we learn a different network for predicting special transforms *for each drawing command* (value of $t_1$) and also *for each token of the drawing command*. These networks ($\mathrm{MLP}_{t_1,n}$ in equation 5) have no hidden layers and output the 6 entries of an affine transformation matrix; see [7] for more details.

Training takes a little bit less than a day on a Nvidia TitanX GPU. The network was trained on $10^5$ synthetic examples.

## 1.4 LSTM Baseline

We compared our deep network with a baseline that models the problem as a kind of image captioning. Given the target image, this baseline produces the program spec in one shot by using a CNN to extract features of the input which are passed to an LSTM which finally predicts the spec token-by-token. This general architecture is used in several successful neural models of image captioning (e.g., [8]).

Concretely, we kept the image feature extractor architecture (a CNN) as in our model, but only passed it one image as input (the target image to explain). Then, instead of using an autoregressive decoder to predict a single drawing command, we used an LSTM to predict a sequence of drawing commands token-by-token. This LSTM had 128 memory cells, and at each time step produced as output the next token in the sequence of drawing commands. It took as input both the image representation and its previously predicted token.

Figure 2: Example synthetic training data

## 1.5 Generating synthetic training data

We generated synthetic training data for the neural network by sampling LaTeX code according to the following generative process: First, the number of objects in the scene are sampled uniformly from 1 to 12. For each object we uniformly sample its identity (circle, rectangle, or line). Then we sample the parameters of the circles, than the parameters of the rectangles, and finally the parameters of the lines; this has the effect of teaching the network to first draw the circles in the scene, then the rectangles, and finally the lines. We furthermore put the circle (respectively, rectangle and line) drawing commands in order by left-to-right, bottom-to-top; thus the training data enforces a canonical order in which to draw any scene.

To make the training data look more like naturally occurring figures, we put a Chinese restaurant process prior [9] over the values of the X and Y coordinates that occur in the execution spec. This encourages reuse of coordinate values, and so produces training data that tends to have parts that are nicely aligned.

In the synthetic training data we excluded any sampled scenes that had overlapping drawing commands. As shown in the main paper, the network is then able to generalize to scenes with, for example, intersecting lines or lines that penetrate a rectangle.

When sampling the endpoints of a line, we biased the sampling process so that it would be more likely to start an endpoint along one of the sides of a rectangle or at the boundary of a circle. If $n$ is the number of points either along the side of a rectangle or at the boundary of a circle, we would sample an arbitrary endpoint with probability $\frac{2}{2+n}$ and sample one of the "attaching" endpoints with probability $\frac{1}{2+n}$.

See figure 2 for examples of the kinds of scenes that the network is trained on.

For readers wishing to generate their own synthetic training sets, we refer them to our source code at: `https://github.com/ellisk42/TikZ`.

## 2 Generalizing to real hand drawings

### 2.1 Simulating hand drawings

We introduce noise into the LaTeX rendering process by:

- Rescaling the image intensity by a factor chosen uniformly at random from $[0.5, 1.5]$

- Translating the image by $\pm 3$ pixels chosen uniformly random

- Rendering the LaTeX using the `pencildraw` style, which adds random perturbations to the paths drawn by LaTeX in a way designed to resemble a pencil.

- Randomly perturbing the positions and sizes of primitive LaTeX drawing commands

hand drawing

rendering of (a)'s
inferred spec

noisy rendering
of (b)

Figure 3: Noisy renderings produced in LaTeX TikZ w/ `pencildraw` package

## 2.2 A learned likelihood surrogate

Our architecture for $L_{\text{learned}}(\text{render}(T_1)|\text{render}(T_2))$ has the same series of convolutions as the network that predicts the next drawing command. We train it to predict two scalars: $|T_1 - T_2|$ and $|T_2 - T_1|$. These predictions are made using linear regression from the image features followed by a ReLU nonlinearity; this nonlinearity makes sense because the predictions can never be negative but could be arbitrarily large positive numbers.

We train this network by sampling random synthetic scenes for $T_1$, and then perturbing them in small ways to produce $T_2$. We minimize the squared loss between the network's prediction and the ground truth symmetric differences. $T_1$ is rendered in a "simulated hand drawing" style which Section 2.1 describes.

## 3 The cost function for programs

We seek the minimum cost program which evaluates to (produces the drawing primitives in) an execution spec $T$:

$$\text{program}(T) = \underset{\substack{p \in \text{DSL} \\ p \text{ evaluates to } T}}{\arg\min} \quad \text{cost}(p) \tag{6}$$

Programs incur a cost of 1 for each command (primitive drawing action, loop, or reflection). They incur a cost of $\frac{1}{3}$ for each unique coefficient they use in a linear transformation beyond the first coefficient. This encourages reuse of coefficients, which leads to code that has translational symmetry; rather than provide a translational symmetry operator as we did with reflection, we modify what is effectively a prior over the space of program so that it tends to produce programs that have this symmetry.

Programs also incur a cost of 1 for having loops of constant length 2; otherwise there is often no pressure from the cost function to explain a repetition of length 2 as being a reflection rather a loop.

## 4 Learning a search policy

Figure 4 provides additional intuition for how the policy biases and informs the program synthesizer: the entire program search space is carved up into smaller subsets, and we search within each of these subsets *simultaneously and in parallel*, but where the fraction of compute time allocated to each subset is proportional to the weight assigned to it by the policy.

Figure 4: The bias-optimal search algorithm divides the entire (intractable) program search space in to (tractable) program subspaces (written $\sigma$), each of which contains a restricted set of programs. For example, one subspace might be short programs which don't loop. The policy $\pi$ predicts a distribution over program subspaces. The weight that $\pi$ assigns to a subspace is indicated by its yellow shading in the above figure, and is conditioned on the spec $S$.

## 4.1 Modeling

Recall from the main paper that our goal is to estimate the policy minimizing the following loss:

$$\text{LOSS}(\theta; \mathcal{D}) = \mathbb{E}_{S \sim \mathcal{D}}\left[\min_{\sigma \in \text{BEST}(S)} \frac{t(\sigma|S)}{\pi_\theta(\sigma|S)}\right] + \lambda\|\theta\|_2^2 \tag{7}$$

$$\text{where } \sigma \in \text{BEST}(S) \text{ if a minimum cost program for } S \text{ is in } \sigma.$$

We make this optimization problem tractable by annealing our loss function during gradient descent:

$$\text{LOSS}_\beta(\theta; \mathcal{D}) = \mathbb{E}_{S \sim \mathcal{D}}\left[\text{SOFTMINIMUM}_\beta\left\{\frac{t(\sigma|S)}{\pi_\theta(\sigma|S)} : \sigma \in \text{BEST}(S)\right\}\right] + \lambda\|\theta\|_2^2 \tag{8}$$

$$\text{where SOFTMINIMUM}_\beta(x_1, x_2, x_3, \cdots) = \sum_n x_n \frac{e^{-\beta x_n}}{\sum_{n'} e^{-\beta x_{n'}}} \tag{9}$$

Notice that $\text{SOFTMINIMUM}_{\beta=\infty}(\cdot)$ is just $\min(\cdot)$. We set the regularization coefficient $\lambda = 0.1$ and minimize equation 8 using Adam for 2000 steps, linearly increasing $\beta$ from 1 to 2.

We parameterize the space of policies as a simple log bilinear model:

$$\pi_\theta(\sigma|S) \propto \exp\left(\phi_{\text{params}}(\sigma)^\top \theta \phi_{\text{spec}}(S)\right) \tag{10}$$

where:

$$\phi_{\text{params}}(\sigma) = [\mathbb{1}[\sigma \text{ can loop}];$$
$$\mathbb{1}[\sigma \text{ can reflect}];$$
$$\mathbb{1}[\sigma \text{ is incremental}];$$
$$\mathbb{1}[\sigma \text{ has depth bound 1}]; \mathbb{1}[\sigma \text{ has depth bound 2}]; \mathbb{1}[\sigma \text{ has depth bound 3}];]$$
$$\phi_{\text{spec}}(S) = [\# \text{ circles in } S; \# \text{ rectangles in } S; \# \text{ lines in } S; 1]$$

where the meaning of "incremental" is described in the next section.

## 4.2 Incremental Solving

Rather than give the sketch program synthesizer the entire spec all at once, we can instead give it subsets of the spec (subsets of the objects in the image) and ask it to synthesize a program for each subset. We then concatenate the resulting programs from each subset to get a program that explains the entire image, and we call this strategy "incremental solving". Incremental solving is

not guaranteed to be faster, nor is it guaranteed to find a minimum cost program. Thus we allow the search policy to decide what fraction of our search time should be allocated to this incremental approach to program synthesis. Concretely, we partitioned a spec into its constituent lines, circles, and rectangles.

### 4.3  Baseline comparisons

#### 4.3.1  DeepCoder-style baseline

In addition to the end-to-end baseline, we compared with a DeepCoder-style baseline (main paper, Section 3.1). DeepCoder (DC) [5] is an approach for learning to speed up program synthesizers. DC models are neural networks that predict, starting from a spec, the probability of a DSL component being in a minimal-cost program satisfying the spec. Writing $\text{DC}(S)$ for the distribution predicted by the neural network, DC is trained to maximize the following objective:

$$\mathbb{E}_{S \sim \mathcal{D}} \left[ \min_{p \in \text{BEST}(S)} \sum_{x \in \text{DSL}} \log \left( \mathbb{1}\left[ x \in p \right] \text{DC}(S)_x + \mathbb{1}\left[ x \notin p \right] \left( 1 - \text{DC}(S)_x \right) \right) \right] \tag{11}$$

where $x$ ranges over DSL components and $\text{DC}(S)_x \in [0, 1]$ is the probability predicted by the DC model for component $x$ for spec $S$.

We provided our DC model with the same features given to our bias optimal search policy ($\phi_{spec}$ in section 4.1), and trained using the same 20-fold cross validation splits. To evaluate the DC baseline on held out data, we used the *Sort-and-Add* policy described in the DeepCoder paper[5].

#### 4.3.2  End-to-End baseline

Recall that we factored the graphics program synthesis problem into two components: (1) a perception-facing component, whose job is to go from perceptual input to a set of commands that must occur in the execution of the program (**spec**); and (2) a program synthesis component, whose job is to infer a program whose execution contains those commands. This is a different approach from other recent program induction models (e.g., [1, 2]), which regress directly from a program induction problem to the source code of the program.

**Experiment.** To test whether this factoring is necessary for our domain, we trained a model to regress directly from images to graphics programs. This baseline model, which we call the *no-spec baseline*, was able to infer some simple programs, but failed completely on more sophisticated scenes.

Baseline model architecture: The model architecture is a straightforward, image-captioning-style CNN→LSTM. We keep the same CNN architecture from our main model (Section 1.2), with the sole difference that it takes only one image as input. The LSTM decoder produces the program token-by-token: so we flatten the program's hierarchical structure, and use special "bracketing" symbols to convey nesting structure, in the spirit of [3]. The LSTM decoder has 2 hidden layers with 1024 units. We used 64-dimensional embeddings for the program tokens.

Training and evaluation: The model was trained on $10^7$ synthetically generated programs – 2 orders of magnitude more data than the model we present in the main paper. We then evaluated the baseline on *synthetic renders* of our 100 hand drawings (the testing set used throughout the paper). Recall that our model was evaluated on noisy real hand drawings. We sample programs from this baseline model conditioned on a synthetic render of a hand drawing, and report only the sampled program whose output most closely matched the ground truth spec spec, as measured by the symmetric difference of the two sets. We allow the baseline model to spend 1 hour drawing samples per drawing – recall that our model finds 58% of programs in under a minute. Together these training and evaluation choices are intended to make the problem as easy as possible for the baseline.

Results: The no-spec baseline succeeds for trivial programs (a few lines, no variables, loops, etc.); occasionally gets small amounts of simple looping structure; and fails utterly for most of our test cases. See Figure 5.

Figure 5: Top, white: synthetic rendering of a hand drawing. Bottom, black: output of best program found by no-spec baseline.

# 5 Correcting errors made by the neural network

The program synthesizer can help correct errors from the execution spec proposal network by favoring execution specs which lead to more concise or general programs. For example, one generally prefers figures with perfectly aligned objects over figures whose parts are slightly misaligned – and precise alignment lends itself to short programs. Similarly, figures often have repeated parts, which the program synthesizer might be able to model as a loop or reflectional symmetry. So, in considering several candidate specs proposed by the neural network, we might prefer specs whose best programs have desirable features such being short or having iterated structures.

Concretely, we implemented the following scheme: for an image $I$, the neurally guided sampling scheme of section 2 of the main paper samples a set of candidate specs, written $\mathcal{F}(I)$. Instead of predicting the most likely spec in $\mathcal{F}(I)$ according to the neural network, we can take into account the programs that best explain the specs. Writing $\hat{S}(I)$ for the spec the model predicts for image $I$,

$$\hat{S}(I) = \arg\max_{S \in \mathcal{F}(I)} L_{\text{learned}}(I|\text{render}(S)) \times \mathbb{P}_\theta[S|I] \times \mathbb{P}_\beta[\text{program}(S)] \qquad (12)$$

where $\mathbb{P}_\beta[\cdot]$ is a prior probability distribution over programs parameterized by $\beta$. This is equivalent to doing MAP inference in a generative model where the program is first drawn from $\mathbb{P}_\beta[\cdot]$, then the program is executed deterministically, and then we observe a noisy version of the program's output, where $L_{\text{learned}}(I|\text{render}(\cdot)) \times \mathbb{P}_\theta[\cdot|I]$ is our observation model.

Given a corpus of graphics program synthesis problems with annotated ground truth specs (i.e. $(I, S)$ pairs), we find a maximum likelihood estimate of $\beta$:

$$\beta^* = \arg\max_\beta \mathbb{E}\left[\log \frac{\mathbb{P}_\beta[\text{program}(S)] \times L_{\text{learned}}(I|\text{render}(S)) \times \mathbb{P}_\theta[S|I]}{\sum_{S' \in \mathcal{F}(I)} \mathbb{P}_\beta[\text{program}(S')] \times L_{\text{learned}}(I|\text{render}(S')) \times \mathbb{P}_\theta[S'|I]}\right] \qquad (13)$$

where the expectation is taken both over the model predictions and the $(I, S)$ pairs in the training corpus. We define $\mathbb{P}_\beta[\cdot]$ to be a log linear distribution $\propto \exp(\beta \cdot \phi(\text{program}))$, where $\phi(\cdot)$ is a feature extractor for programs. We extract a few basic features of a program, such as its size and how many loops it has, and use these features to help predict whether a spec is the correct explanation for an image.

We synthesized programs for the top 10 specs output by the deep network. Learning this prior over programs can help correct mistakes made by the neural network, and also occasionally introduces mistakes of its own; see Fig. 6 for a representative example of the kinds of corrections that it makes. On the whole it modestly improves our Top-1 accuracy from 63% to 67%. Recall that from Fig. 6 of the main paper that the best improvement in accuracy we could possibly get is 70% by looking at the top 10 specs.

# 6 Measuring similarity between drawings

We measure the similarity between two drawings by extracting features of the best programs that describe them. Our features are counts of the number of times that different components in the DSL

Figure 6: Left: hand drawing. Center: interpretation favored by the deep network. Right: interpretation favored after learning a prior over programs. Our learned prior favors shorter, simpler programs, thus (top example) continuing the pattern of not having an arrow is preferred, or (bottom example) continuing the binary search tree is preferred.

were used. We project these features down to a 2-dimensional subspace using primary component analysis (PCA); see Fig.7. One could use many alternative similarity metrics between drawings which would capture pixel-level similarities while missing high-level geometric similarities. We used our learned distance metric between specs, $L_{\text{learned}}(\cdot|\cdot)$, and projected to a 2-dimensional subspace using multidimensional scaling (MDS: [4]). This reveals similarities between the objects in the drawings, while missing similarities at the level of the program.

## 7 Full results on drawings data set

Below we show our full data set of drawings. The leftmost column is a hand drawing. The middle column is a rendering of the most likely spec discovered by the neurally guided SMC sampling scheme. The rightmost column is the program we synthesized from a ground truth execution spec of the drawing. Note that because the inference procedure is stochastic, the top one most likely sample can vary from run to run. Below we report a representative sample from a run with 2000 particles.

```
line(6,2,6,3,
arrow = False,solid = True);
line(6,2,3,2,
arrow = True,solid = True);
reflect(y = 9){
line(3,7,5,5,
arrow = True,solid = True);
rectangle(1,1,3,3);
rectangle(5,3,7,6);
rectangle(0,0,8,9)
}
```

Figure 7: PCA on features of the programs that were synthesized for each drawing. Symmetric figures cluster to the right; "loopy" figures cluster to the left; complicated programs are at the top and simple programs are at the bottom.

Figure 8: MDS on drawings using the learned distance metric, $L_{\text{learned}}(\cdot|\cdot)$. Drawings with similar looking parts in similar locations are clustered together.

```
for (i < 2){
line(8,8,3,8,
arrow = True,solid = False);
line(-2 * i + 12,5,-2 * i + 13,5
arrow = True,solid = True);
line(6,5,7,5,
arrow = True,solid = True);
line(3,-6 * i + 8,5,-2 * i + 6,
arrow = True,solid = True);
rectangle(-2 * i + 13,4,-2 * i +
rectangle(1,-6 * i + 7,3,-6 * i
};
circle(8,5);
rectangle(5,3,6,7);
rectangle(0,0,10,10);
```

```
reflect(y = 7){
line(2,6,4,4,
arrow = True,solid = True);
rectangle(0,0,2,2)
};
rectangle(4,2,6,5)
```

```
line(7,5,9,5,
arrow = True,solid = True);
rectangle(5,3,7,7);
rectangle(0,0,12,10);
reflect(y = 10){
circle(10,5);
line(3,2,5,4,
arrow = True,solid = True);
rectangle(1,1,3,3)
}
```

```
line(10,1,2,1,
arrow = True,solid = False);
line(10,1,10,3,
arrow = False,solid = False);
line(7,4,9,4,
arrow = True,solid = True);
reflect(y = 8){
circle(10,4);
line(2,1,4,3,
arrow = True,solid = True);
rectangle(4,2,7,6);
rectangle(0,6,2,8)
}
```

```
line(12,9,12,0,
arrow = True,solid = True);
rectangle(9,3,11,9);
rectangle(6,5,8,9);
rectangle(0,7,2,9);
rectangle(3,8,5,9)
```

```
for (i < 3){
for (j < (1*i + 1)){
if (j > 0){
line(3 * j + -3,3 * i + -2,3 * j
arrow = False,solid = True);
line(0,3 * j + -2,3 * j + -3,4,
arrow = False,solid = True)
}
rectangle(2,0,5,3)
}
}
```

```
for (i < 3){
circle(-3 * i + 7,1);
circle(-3 * i + 7,6);
line(-3 * i + 7,-1 * i + 4,-3 *
arrow = False,solid = True)
}
```

```
line(0,0,0,4,
arrow = False,solid = True)
```

```
line(6,0,0,0,
arrow = True,solid = True)
```

```
rectangle(0,0,3,4)
```

```
circle(1,1)
```

```
reflect(x = 7){
circle(6,1);
line(6,2,6,5,
arrow = False,solid = True);
rectangle(5,5,7,7)
};
line(2,6,5,6,
arrow = False,solid = True);
line(2,1,5,1,
arrow = False,solid = True)
```

```
line(3,2,1,2,
arrow = True,solid = True);
line(0,3,2,3,
arrow = True,solid = True);
line(5,0,3,0,
arrow = True,solid = True);
line(2,1,4,1,
arrow = True,solid = True)
```

```
rectangle(6,0,7,1);
for (i < 3){
rectangle(-2 * i + 4,2 * i + 2,-
rectangle(-2 * i + 4,2 * i,-2 *
}
```

```
line(3,0,5,0,
arrow = False,solid = True);
line(1,2,3,2,
arrow = False,solid = True);
line(0,3,2,3,
arrow = False,solid = False);
line(2,1,4,1,
arrow = False,solid = False)
```

```
circle(9,1);
for (i < 3){
circle(-2 * i + 7,3 * i + 4);
circle(-2 * i + 5,3 * i + 1);
line(-2 * i + 6,3 * i + 1,-2 * i
arrow = False,solid = True);
line(-2 * i + 7,3 * i + 3,-2 * i
arrow = False,solid = True)
}
```

```
for (i < 3){
line(2 * i + 3,-3 * i + 9,2 * i
arrow = True,solid = True);
line(2 * i + 3,-3 * i + 9,2 * i
arrow = True,solid = True);
rectangle(2 * i + 2,-3 * i + 9,2
rectangle(2 * i,-3 * i + 6,2 * i
};
rectangle(8,0,10,2)
```

```
for (i < 2){
circle(2 * i + 1,-3 * i + 6);
circle(-3 * i + 7,2 * i + 3);
circle(2 * i + 5,1)
}
```

```
line(4,4,2,2,
arrow = True,solid = True);
rectangle(0,0,2,2);
rectangle(3,4,5,6)
```

```
for (i < 3){
line(-4 * i + 9,4,-4 * i + 9,2,
arrow = True,solid = True);
for (j < (1*i + 2)){
if (j > 0){
circle(-4 * j + 13,-4 * i + 9);
line(-4 * i + 12,5,-4 * i + 10,5
arrow = True,solid = True)
}
rectangle(0,4,2,6)
}
}
```

```
circle(1,5);
line(4,1,2,1,
arrow = True,solid = True);
line(8,1,6,1,
arrow = True,solid = True);
for (i < 3){
line(4 * i + 1,2,4 * i + 1,4,
arrow = True,solid = True);
rectangle(4,4,6,6);
rectangle(4 * i,0,4 * i + 2,2)
};
rectangle(8,4,10,6)
```

```
for (i < 3){
line(4 * i + 1,4,4 * i + 1,2,
arrow = True,solid = True);
for (j < 2){
line(4 * j + 4,5,4 * j + 2,5,
arrow = True,solid = True);
rectangle(4 * i,-4 * j + 4,4 * i
}
}
```

```
for (i < 3){
line(4 * i + 1,2,4 * i + 1,4,
arrow = True,solid = True);
for (j < 2){
circle(4 * i + 1,4 * j + 1);
line(4 * j + 4,1,4 * j + 2,1,
arrow = True,solid = True)
}
}
```

```
line(5,7,5,6,
arrow = True,solid = True);
line(3,3,3,2,
arrow = True,solid = True);
line(1,7,1,6,
arrow = True,solid = True);
rectangle(0,3,6,6);
rectangle(2,0,4,2);
rectangle(0,7,6,9)
```

```
line(6,1,5,1,
arrow = True,solid = True);
for (i < 3){
line(3,1,2,1,
arrow = True,solid = True);
rectangle(3 * i,0,3 * i + 2,2)
}
```

```
for (i < 3){
circle(1,-4 * i + 9)
};
line(1,4,1,2,
arrow = True,solid = True);
line(1,8,1,6,
arrow = True,solid = True)
```

```
reflect(y = 2){
line(0,1,1,2,
arrow = False,solid = True);
line(1,0,2,1,
arrow = False,solid = True)
}
```

```
line(0,0,0,2,
arrow = False,solid = True);
line(0,2,2,2,
arrow = False,solid = True)
```

```
for (i < 3){
line(1 * i,-2 * i + 4,1 * i,-1 *
arrow = False,solid = True);
line(1 * i,-1 * i + 6,2 * i + 2,
arrow = False,solid = True)
}
```

```
circle(5,2);
circle(5,4);
rectangle(4,1,6,5);
reflect(y = 5){
circle(1,4);
rectangle(0,3,2,5)
}
```

```
for (i < 3){
for (j < (1*i + 1)){
circle(3 * i + 1,-2 * j + 5)
};
rectangle(3 * i,-2 * i + 4,3 * i
}
```

```
circle(5,5);
line(2,5,4,5,
arrow = False,solid = True);
rectangle(0,0,5,3);
rectangle(0,4,2,6)
```

```
line(0,0,0,3,
arrow = False,solid = True);
line(6,0,6,3,
arrow = False,solid = True);
line(0,3,6,3,
arrow = False,solid = False);
line(0,0,6,0,
arrow = False,solid = False)
```

```
for (i < 2){
line(2 * i,-2 * i + 5,2,5,
arrow = False,solid = True);
line(1,2,2 * i + 1,-2 * i + 4,
arrow = False,solid = True);
line(2 * i + 3,-2 * i + 3,5,3,
arrow = False,solid = True);
line(4,0,2 * i + 4,-2 * i + 2,
arrow = False,solid = True)
}
```

```
circle(6,1);
circle(1,1);
circle(1,6);
line(2,6,6,2,
arrow = True,solid = True);
line(1,5,1,2,
arrow = True,solid = True)
```

```
rectangle(5,5,9,9);
rectangle(0,0,4,4);
for (i < 2){
rectangle(1 * i + 7,-2 * i + 2,9
rectangle(0,-3 * i + 8,1 * i + 1
}
```

```
circle(1,8);
line(5,2,5,5,
arrow = False,solid = True);
line(1,7,3,5,
arrow = False,solid = True);
rectangle(4,0,6,2);
rectangle(0,5,6,9)
```

```
for (i < 3){
for (j < 3){
if (j > 0){
line(3 * j + -1,3 * i + 1,3 * j,
arrow = False,solid = True);
line(3 * i + 1,3 * j + -1,3 * i
arrow = False,solid = True)
}
circle(3 * i + 1,3 * j + 1)
}
}
```

```
for (i < 3){
for (j < 3){
if (j > 0){
line(3 * i + 1,3 * j + -1,3 * i
arrow = False,solid = True);
line(3 * j + -1,3 * i + 1,3 * j,
arrow = False,solid = True)
}
rectangle(3 * i,3 * j,3 * i + 2,
}
}
```

```
for (i < 3){
circle(-3 * i + 7,1)
}
```

```
for (i < 3){
rectangle(-2 * i + 4,0,-2 * i +
}
```

```
line(4,0,4,1,
arrow = False,solid = False);
line(0,0,0,5,
arrow = False,solid = False);
line(4,1,4,5,
arrow = False,solid = False)
```

```
line(0,0,0,5,
arrow = False,solid = True);
line(4,0,4,5,
arrow = False,solid = True)
```

```
reflect(x = 12){
circle(4,1);
line(2,1,3,1,
arrow = False,solid = True);
rectangle(0,0,2,2)
}
```

```
circle(7,6);
reflect(y = 12){
line(4,6,6,6,
arrow = True,solid = True);
line(2,10,2,8,
arrow = True,solid = True);
rectangle(1,0,3,2)
};
rectangle(0,4,4,8)
```

```
reflect(y = 9){
reflect(x = 9){
circle(8,8);
line(3,8,6,8,
arrow = False,solid = True);
line(1,3,1,6,
arrow = False,solid = True)
}
}
```

```
reflect(x = 11){
rectangle(9,4,10,7);
reflect(y = 11){
rectangle(8,0,11,3);
rectangle(4,9,7,10)
}
}
```

```
for (i < 3){
line(2 * i,-2 * i + 5,2 * i + 2,
arrow = False,solid = True);
line(2 * i + 1,-2 * i + 4,2 * i
arrow = False,solid = True)
}
```

```
rectangle(4,1,6,2);
rectangle(7,0,9,2);
reflect(y = 10){
rectangle(0,0,3,3);
rectangle(1,4,2,6)
}
```

```
line(7,4,9,4,
arrow = True,solid = True);
line(8,3,7,3,
arrow = True,solid = True);
reflect(y = 7){
line(2,1,4,3,
arrow = True,solid = True);
rectangle(0,0,2,2)
};
line(8,3,9,3,
arrow = False,solid = True);
rectangle(9,2,12,5);
rectangle(4,2,7,5)
```

```
for (i < 3){
rectangle(2 * i,2 * i,2 * i + 3,
}
```

```
circle(4,10);
for (i < 3){
circle(3 * i + 1,1);
circle(3 * i + 1,5);
line(4,9,3 * i + 1,6,
arrow = True,solid = True);
line(3 * i + 1,4,3 * i + 1,2,
arrow = True,solid = True)
}
```

```
line(2,8,2,6,
arrow = True,solid = True);
line(4,8,4,0,
arrow = True,solid = True);
line(6,8,6,4,
arrow = True,solid = True);
line(0,8,8,8,
arrow = False,solid = True)
```

```
line(2,3,2,5,
arrow = False,solid = True);
rectangle(0,0,4,8);
rectangle(1,1,3,3);
rectangle(1,5,3,7)
```

```
circle(1,5);
line(1,4,1,2,
arrow = True,solid = True);
rectangle(0,0,2,2)
```

```
rectangle(0,0,6,2);
reflect(x = 6){
for (i < 3){
circle(5,2 * i + 4);
circle(2 * i + 1,1);
rectangle(4,3,6,9)
}
}
```

```
for (i < 3){
for (j < 3){
circle(4 * i + 1,-3 * j + 7)
}
}
```

```
line(8,0,0,0,
arrow = True,solid = True);
line(8,0,8,7,
arrow = True,solid = True);
for (i < 3){
rectangle(-2 * i + 6,0,-2 * i +
}
```

```
line(4,0,0,0,
arrow = False,solid = False)
```

```
line(2,4,4,4,
arrow = True,solid = True);
line(6,7,5,5,
arrow = True,solid = True);
for (i < 2){
circle(-2 * i + 7,-3 * i + 7);
circle(3 * i + 4,1);
line(5,3,3 * i + 4,2,
arrow = True,solid = True);
rectangle(0,3,2,5)
}
```

```
circle(2,1);
circle(6,1);
line(5,1,3,1,
arrow = True,solid = True);
rectangle(0,0,7,2)
```

```
rectangle(5,0,8,3);
rectangle(0,2,1,3);
rectangle(2,1,4,3)
```

```
for (i < 3){
rectangle(-1 * i + 2,-1 * i + 2,
}
```

```
reflect(y = 6){
line(2,5,4,5,
arrow = False,solid = True);
reflect(x = 6){
line(5,2,5,4,
arrow = False,solid = True);
rectangle(0,4,2,6)
}
}
```

```
reflect(y = 6){
reflect(x = 6){
circle(1,1);
line(5,2,5,4,
arrow = False,solid = True)
};
line(2,1,4,1,
arrow = False,solid = True)
}
```

```
for (i < 3){
line(1 * i,-1 * i + 2,-1 * i + 7
arrow = False,solid = True)
}
```

```
line(1,5,5,1,
arrow = False,solid = True);
line(1,4,5,0,
arrow = False,solid = True);
rectangle(0,4,1,5);
rectangle(5,0,6,1)
```

```
for (i < 3){
circle(-4 * i + 9,1);
rectangle(-4 * i + 8,0,-4 * i +
}
```

```
reflect(x = 5){
circle(4,1);
line(4,4,4,2,
arrow = True,solid = True)
};
rectangle(0,4,5,6)
```

```
circle(3,1);
reflect(x = 6){
circle(5,5);
circle(1,9);
line(5,4,3,2,
arrow = True,solid = True);
line(5,8,2,5,
arrow = True,solid = True);
line(1,8,1,6,
arrow = True,solid = True)
}
```

```
for (i < 3){
line(7,1,5 * i + 2,3,
arrow = True,solid = True);
for (j < (1*i + 1)){
if (j > 0){
line(5 * j + -1,9,5 * i,5,
arrow = True,solid = True)
}
line(5 * j + 2,5,5 * j + 2,9,
arrow = True,solid = True)
};
rectangle(5 * i,3,5 * i + 4,5);
rectangle(5 * i,9,5 * i + 4,10)
};
rectangle(2,0,12,1)
```

```
reflect(y = 8){
for (i < 3){
circle(-3 * i + 7,-3 * i + 7)
};
rectangle(2,2,3,3);
rectangle(5,5,6,6)
}
```

```
line(10,8,12,4,
arrow = True,solid = True);
line(6,8,8,4,
arrow = True,solid = True);
for (i < 3){
line(4 * i + 5,5,4 * i + 5,7,
arrow = True,solid = True);
line(4 * i + 5,1,4 * i + 5,3,
arrow = True,solid = True);
rectangle(4 * i + 4,3,4 * i + 6,
rectangle(4 * i + 4,7,4 * i + 6,
line(2,1,13,1,
arrow = False,solid = True)
};
rectangle(0,0,2,8)
```

```
for (i < 3){
for (j < 3){
circle(2 * j + 1,2 * i + 1)
}
};
rectangle(2,2,4,4);
rectangle(0,0,6,6)
```

```
for (i < 4){
circle(4 * i + 1,1);
circle(4 * i + 1,5);
for (j < 3){
line(4 * i + 1,4,4 * i + 1,2,
arrow = True,solid = True);
line(4 * j + 2,5,4 * j + 4,5,
arrow = True,solid = True)
}
}
```

```
reflect(x = 8){
circle(4,1);
circle(1,8);
line(0,2,4,5,
arrow = False,solid = True);
line(4,5,4,10,
arrow = False,solid = True)
}
```

```
circle(9,8);
circle(5,1);
circle(1,8);
reflect(x = 10){
line(6,1,9,3,
arrow = False,solid = True);
line(2,8,4,8,
arrow = False,solid = True);
line(9,5,9,7,
arrow = False,solid = True);
rectangle(0,3,2,5)
};
rectangle(4,7,6,9)
```

```
line(3,2,5,4,
arrow = True,solid = True);
line(6,6,6,5,
arrow = True,solid = True);
line(8,3,7,4,
arrow = True,solid = True);
line(4,0,12,8,
arrow = False,solid = True);
line(0,6,12,6,
arrow = False,solid = True);
line(0,8,8,0,
arrow = False,solid = True)
```

```
for (i < 3){
circle(4 * i + 1,13);
circle(5,-4 * i + 9);
circle(4 * i + 1,9);
line(4 * i + 1,12,4 * i + 1,10,
arrow = True,solid = True);
line(5,-4 * i + 12,5,-4 * i + 10
arrow = True,solid = True)
};
line(9,8,6,5,
arrow = True,solid = True);
line(1,8,4,5,
arrow = True,solid = True)
```

```
reflect(x = 14){
circle(11,10);
circle(3,4);
circle(7,1);
reflect(y = 20){
circle(13,7);
circle(9,7)
};
line(3,3,7,2,
arrow = True,solid = True);
line(10,10,5,8,
arrow = True,solid = True);
reflect(x = 6){
line(5,12,3,11,
arrow = True,solid = True);
line(1,6,3,5,
arrow = True,solid = True);
line(3,9,5,8,
arrow = True,solid = True)
}
}
```

```
circle(1,1);
circle(4,1);
rectangle(6,0,8,2);
rectangle(9,0,11,2)
```

Solver timeout

```
reflect(x = 10){
circle(5,1);
circle(2,4);
line(2,3,5,2,
arrow = True,solid = True);
reflect(x = 16){
circle(9,7);
line(9,6,8,5,
arrow = True,solid = True)
}
}
```

Solver timeout

```
for (i < 4){
if (i > 0){
line(1,3 * i,1,3 * i + -1,
arrow = True,solid = True)
}
circle(1,3 * i + 1)
}
```

```
rectangle(5,0,8,3);
rectangle(0,5,3,8);
for (i < 2){
rectangle(1 * i + 6,3 * i + 4,8,
rectangle(0,2 * i,1 * i + 1,3 *
}
```

```
reflect(x = 8){
rectangle(0,0,1,1);
rectangle(5,5,8,8);
rectangle(0,2,2,4)
}
```

```
for (i < 3){
rectangle(-2 * i + 4,-1 * i + 2,
}
```

```
circle(9,5);
line(3,1,8,1,
arrow = True,solid = True);
line(8,5,7,5,
arrow = True,solid = True);
line(9,8,0,8,
arrow = True,solid = True);
line(9,2,9,4,
arrow = True,solid = True);
line(12,1,10,1,
arrow = True,solid = True);
line(9,2,10,1,
arrow = False,solid = True);
line(12,1,12,5,
arrow = False,solid = True);
reflect(x = 6){
line(6,5,4,5,
arrow = True,solid = True)
};
rectangle(2,4,4,6);
rectangle(1,0,13,9);
line(9,6,9,8,
arrow = False,solid = True);
line(6,4,7,5,
arrow = False,solid = True);
line(10,5,12,5,
arrow = False,solid = True);
line(3,1,3,4,
arrow = False,solid = True)
```

```
circle(6,2);
for (i < 3){
circle(5 * i + 1,7)
};
line(5,7,2,7,
arrow = True,solid = True);
line(6,6,6,3,
arrow = True,solid = True);
line(10,7,7,7,
arrow = True,solid = True);
rectangle(4,0,8,9)
```

```
reflect(x = 5){
reflect(y = 5){
line(0,5,2,5,
arrow = False,solid = True);
line(0,3,0,5,
arrow = False,solid = True)
}
}
```

```
for (i < 3){
reflect(x = 14){
circle(9,4 * i + 1);
line(10,4 * i + 1,12,4 * i + 1,
arrow = False,solid = True);
rectangle(0,4 * i,2,4 * i + 2)
}
}
```

```
reflect(x = 10){
line(5,6,1,8,
arrow = True,solid = True);
line(9,2,5,4,
arrow = True,solid = True);
for (i < 3){
circle(4 * i + 1,4 * i + 1)
}
}
```

```
reflect(x = 12){
line(6,2,6,3,
arrow = True,solid = True);
line(2,7,5,4,
arrow = True,solid = True);
line(0,0,9,9,
arrow = False,solid = True)
};
line(0,2,12,2,
arrow = False,solid = True)
```

```
for (i < 3){
for (j < 3){
if (j > 0){
circle(6 * i + -5,-5 * j + 16);
line(6 * i + -5,5,4,2,
arrow = True,solid = True);
line(6 * j + -5,10,6 * i + -5,7,
arrow = True,solid = True)
}
circle(4,1)
}
}
```

```
reflect(y = 10){
circle(1,9);
for (i < 4){
circle(-2 * i + 9,-2 * i + 9)
}
}
```

```
for (i < 3){
circle(1,-4 * i + 9);
circle(5,-4 * i + 9);
for (j < 3){
if (j > 0){
line(4 * i + -3,-4 * j + 10,4 *
arrow = False,solid = True)
}
line(2,-4 * j + 9,4,-4 * j + 9,
arrow = False,solid = True)
}
}
```

```
circle(2,2);
circle(2,6);
circle(2,11);
line(2,5,2,3,
arrow = True,solid = True);
line(2,10,2,7,
arrow = True,solid = True);
rectangle(0,0,4,9)
```

```
for (i < 2){
circle(4,6 * i + 1);
circle(1,6 * i + 4);
rectangle(0,6 * i,2,6 * i + 2);
rectangle(3,6 * i + 3,5,6 * i +
}
```