[Reviews · NeurIPS 2018]

Reviewer 1



The paper presents a system for inferring vector graphics programs from hand drawn raster images. The proposed method first generates a 'spec' (which is conceptually just an unrolled program trace) using a neurally guided sequential monte carlo scheme, and then compresses this spec into a program using the Sketch synthesizer augmented with a neural policy. I believe that this is a strong submission that should be accepted Novelty: The task presented in the paper seems to not have been considered much in previous literature. One exception is Ganin et al. "Synthesizing Programs for Images using Reinforced Adversarial Learning", which appeared on Arxiv before the NIPS submission date, so should be cited. The paper combines an number of ideas: (1) a novel CNN + MLP architecture for generating grammatically correct specs from images (2) neurally guided SMC sampling of specs (3) a robust learned distance metric between images (4) a learned bias optimal search policy on to of Sketch. This combination of methods is novel and figures illustrate the benefits that each component brings. Significance: Combining the perceptual power of neural networks with the interpretability and extrapolation properties of programs is a very exciting research direction and this paper is a great demonstration of the utility of this combination. I think that the application is of real practical use for technical drawing generation and the extensive results presented in the paper show that the approach is already good enough to be a useful tool. Clarity: Since there are many ideas in this paper (see (1)-(4) above) the presentation is very dense to fit in 8 pages. I had to refer to the appendix to really understand the neural network architecture and search policy training, but I think that with so much content it will be difficult to achieve a better balance between clarity and compactness. One thing that I could not find was a precise definition of "number of errors" in Fig 4.

Reviewer 2



= Summary The paper presents a two-stage method to infer programs that generate input drawings. Initially, a CNN architecture is applied to translate the drawing into a sequence of primitive draw commands (lines, rectangles, circles, etc). Then, a program synthesizer is used to collapse the straight-line program into a proper program with conditionals and loops. First, let me say that I absolutely love the results of this paper. It is a wonderful combination of relatively simple tools to achieve something unexpected. However, the paper itself has a bunch of problems, mainly because it has to cover a lot of content in little space. Maybe 8 pages are just not the right format for it and it should go straight to a journal, or maybe splitting the two contributions into two papers would make sense. I should say that I have seen this paper under submission before (at ICLR'18). I think the paper can be accepted as it is, but its long version (including much of the content in the supplementary material in the main text) would always be easier and more interesting to read. = Quality The results are great and the experiments supporting the evaluation are thoughtfully selected, answering most of my questions. Most of the remaining questions have to do with baselines that appeared after the initial version of this paper appeared. For example, in Sect. 2, a comparison to the very recent SPIRAL [A] is an obvious question. However, as [A] appeared on arxiv in April, this can hardly be expected for a NIPS submission, but it would be an interesting point for a future (final?) version of this paper. In Sect. 3, a comparison to more recent work building on top of DeepCoder (e.g. [B]) would have been interesting, as these approaches use the partial program generated so far to drive the search for the remaining program; something that seems especially helpful in the graphical domain. = Clarity This is by far the weakest point of the paper; many finer points only become clear with the supplement in hand. I understand that this is due to the page limit, but it makes the 8 page submission hard to read. One part that I found unclear (even with supplement) was the precise search method in Sect. 3, i.e., how is the search policy integrated with the backend search? = Originality To the best of my knowledge, both the graphics-to-statements method as well as the biased statements-to-program method are new contributions that pull together ideas from a wide range of different fields. = Significance Can I please have this thing to generate tikz from my awful drawings? The presented task is very cute and could be useful to translate pen inputs to vector graphics or LaTeX. The combination of perceptual inputs and program synthesis shows the way for future integration of the two fields, for example for learning repetitive tasks from demonstration. The extrapolation idea in Sect. 4.2 seems nice, but is probably just a toy. The error correct idea from Sect. 4.1 feels underexplored, and a tighter integration of program synthesis method with the graphical component could further improve on this. References: [A] Yaroslav Ganin, Tejas Kulkarni, Igor Babuschkin, S. M. Ali Eslami and Oriol Vinyals. Synthesizing Programs for Images using Reinforced Adversarial Learning. [B] Ashwin Kalyan, Abhishek Mohta, Alex Polozov, Dhruv Batra, Prateek Jain and Sumit Gulwani. Neural-Guided Deductive Search for Real-Time Program Synthesis from Examples.

Reviewer 3



This paper presents a model for synthesizing Latex code from (hand) drawn diagrams. It does so by combining two approaches: i) a NN-based model for generating graphics primitives that reconstruct the image, and ii) synthesize a program from these generated primitives to achieve a better/compressed/more general representation than just a ‘list’ of graphics primitives. Strengths: - I quite enjoyed reading the paper as the work in it seems original, solid and complete (even a bit more than enough). The conclusions are well supported by concrete (and importantly, relevant) evaluation. Weaknesses: - my only objection to the paper is that it packs up quite a lot of information, and because of the page-limits it doesn’t include all the details necessary to reconstruct the model. This means cuts were made, some of which are not warranted. Sure, the appendix is there, but the reader needs to get all the necessary details in the main body of the paper. I quite enjoyed the paper, I think it’s definitely NIPS material, but it needs some additional polishing. I added my list of suggestions I think would help improve readability of the paper at the end of the review. Questions: - I might have missed the point of section 4.2 - I see it as a (spacewise-costly) way to say “programs (as opposed to specs) are a better choice as they enable generalization/extrapolation via changing variable values“? What is the experiment there? If it’s just to show that by changing variables, one can extrapolate to different images, I would save space on 1/2 of Figure 9 and focus on lacking parts of the paper (190 - extrapolations produced by our system - how did the system produce those extrapolations? was there a human that changed variable values or is there something in the system enabling this?) - What is + in Figure 3? If elementwise addition, please specify that - Figure 4 caption explains why the number N of particles is not the same across models. However, that still doesn’t stop me from wondering whether there is a significant difference in performance in case all models are using the same number of particles. Do you have that information? - Line 54 mentions that the network can ‘derender’ images with beam search. Is beam search used or not? What is the size of the beam? Is beam search used for each of the N particles? - From what I understood, the model does not have access to previously generated commands. Can you confirm that? - The order of (generated) specs is irrelevant for rendering, but it is for the generation process. How do you cope with that? Do you use a particular order when training the model or do you permute the specs? - Table 2 - “;” denotes OR, right? I would personally use the BNF notation here and use “|” - 153 - ‘minimized 3 using gradient descent’ - how did you treat the fact that min is not differentiable? - Table 5 - this is evaluated on which problems exactly? The same 100 on which the policy was trained? - Please provide some DeepCoder -style baseline details - the same MLP structure? Applied to which engine? A search algorithm or Sketch? - I find 152 - 153 unclear - how did you synthesize minimum cost programs for each \sigma ? \sigma represents a space of possible solutions, no? - Please provide more details on how you trained L_learned - what is the dataset you trained it on (randomly selected pairs of images, sampled from the same pool of randomly generated images, with a twist)? How did you evaluate its performance? What is the error of that model? Was it treated as a regression or as a classification task? - Figure 7 introduces IoU. Is that the same IoU used in segmentation? If so, how does that apply here? Do you count the union/intersection of pixels? Please provide a citation where a reader can quickly understand that measure. Suggestions: - full Table 3 is pretty, but it could easily be halved to save space for more important (missing!) details of the paper - the appendix is very bulky and not well structured. If you want to refer to the appendix, I would strongly suggest to refer to sections/subsections, otherwise a reader can easily get lost in finding the details - Section 2.1 starts strong, promising generalization to real hand drawings, but in the first sentence the reader realizes the model is trained on artificial data. Only in line 91 it says that the system is tested on hand-written figures. I would emphasize that from the beginning. - Line 108 - penalize using many different numerical constants - please provide a few examples before pointing to the supplement. - Line 154 - a bit more detail of the bilinear model would be necessary (how low-capacity?) - 175-176 - see supplement for details. You need to provide some details in the body of the paper! I want to get the idea how you model the prior from the paper and not the supplement - Table 5 - there’s a figure in the appendix which seems much more informative than this Table, consider using that one instead The related work is well written, I would just suggest adding pix2code (https://arxiv.org/abs/1705.07962) and SPIRAL (https://arxiv.org/abs/1804.01118) for completeness. UPDATE: I've read the author feedback and the other reviews. We all agree that the paper is dense, but we seem to like it nevertheless. This paper should be accepted, even as is because it's a valuable contribution, but I really hope authors will invest additional effort into clarifying the parts we found lacking.